**communications** engineering

# Feasibility test of per-flight contrail avoidance in commercial aviation
Aaron Sonabend-W [1,6], Carl Elkin [1,6] ✉, Thomas Dean[2], John Dudley[3], Noman Ali[1], Jill Blickstein[4], Erica Brand[1], Brian Broshears[3], Sixing Chen [1], Zebediah Engberg[2], Mark Galyen[3], Scott Geraedts[1], Nita Goyal [1], Rebecca Grenham[4], Ulrike Hager[1], Deborah Hecker[3], Marco Jany[5], Kevin McCloskey[1], Joe Ng[1], Brian Norris[3], Frank Opel[5], Juliet Rothenberg[1], Tharun Sankar[1], Dinesh Sanekommu[1], Aaron Sarna [1], Ole Schütt [1], Marc Shapiro[2], Rachel Soh[1], Christopher Van Arsdale[1] & John C. Platt [1]

Contrails, formed by aircraft engines, are a major component of aviation's impact on anthropogenic climate change. Contrail avoidance is a potential option to mitigate this warming effect, however, uncertainties surrounding operational constraints and accurate formation prediction make it unclear whether it is feasible. Here we address this gap with a feasibility test through a randomized controlled trial of contrail avoidance in commercial aviation at the per-flight level. Predictions for regions prone to contrail formation came from a physics-based simulation model and a machine learning model. Participating pilots made altitude adjustments based on contrail formation predictions for flights assigned to the treatment group. Using satellite-based imagery we observed 64% fewer contrails in these flights relative to the control group flights, a statistically significant reduction (p = 0.0331). Our targeted per-flight intervention allowed the airline to track their expected vs actual fuel usage, we found that there is a 2% increase in fuel per adjusted flight. This study demonstrates that per-flight detectable contrail avoidance is feasible in commercial aviation.

Condensation trails (contrails) are cirrus clouds that form when water vapor in cold humid air at high altitudes, condenses onto soot particles emitted from aircraft engines. Like other cirrus clouds, contrails have a consequential impact on the planet's temperature by absorbing outgoing longwave radiation and reflecting incoming solar radiation. The Intergovernmental Panel on Climate Change estimates that contrails make a substantial contribution to aviation's impact on global warming[1].

The estimated warming effect comes primarily from persistent contrails, which are created by a small fraction of flights and last from ten minutes to over twenty hours[2]. Persistent contrails are formed when planes fly through ice-supersaturated regions (ISSRs), where the relative humidity with respect to ice is greater than 100%, and temperature conditions satisfy the Schmidt-Appleman Criterion[3]. Aircraft do not often fly through these conditions, and analysis of flight paths has shown that only a small percentage of flights (2–13%) would need to make small adjustments to avoid the majority of estimated contrail warming[4–8]. Indeed, simulations of navigational contrail avoidance (the practice of flying to avoid contrail forming regions) have shown that it could be an extremely cost-effective way to reduce anthropogenic climate forcing from aviation[8,9]. ISSR have a typical size of $\approx 150\ km$ horizontally[10] and several hundred meters vertically[11,12], so navigational contrail avoidance primarily involves flying above or below the contrail forming regions.

The practical implementation of contrail avoidance faces several challenges. An important hurdle is accurately predicting contrail likely zones (CLZs), regions where contrails are likely to form and persist long enough to have substantial climate impact. Numerical weather prediction models, while valuable, are subject to inaccuracies when predicting ice-supersaturation at the fine-grained spatial scales relevant for contrail formation[13–16]. Operational constraints within airline systems, including air traffic management, can also limit the seamless implementation of flight adjustments that are possible in simulations. Proving that these prediction and operational difficulties can be overcome is necessary in order to evaluate whether navigational contrail avoidance is a viable climate change mitigation strategy[17]. Furthermore, rigorously confirming that CLZ avoidance translates to reduced contrail formation requires robust methodologies, such as well-designed randomized controlled trials.

To test the feasibility of contrail avoidance by a single commercial airline, we conducted the study with American Airlines. Ten senior pilots

[1]Google Research, Mountain View, CA, USA. [2]Breakthrough Energy, Contrails, Kirkland, WA, USA. [3]American Airlines, Flight Operations, Dallas-Fort Worth, TX, USA. [4]American Airlines, Sustainability, Dallas-Fort Worth, TX, USA. [5]PACE Aerospace Engineering & IT GmbH, Flight Operations, Berlin, Germany. [6]These authors contributed equally: Aaron Sonabend-W, Carl Elkin. ✉e-mail: celkin@google.com

participated in the trial, completing 44 flights between January and June 2023. Half of the flights (22) were rerouted to avoid CLZs, while the other half kept to their planned routes through CLZs, providing an effective control. We focused exclusively on demonstrating the feasibility of contrail avoidance on a per-flight basis and did not consider the radiative forcing of avoided contrails.

To forecast CLZs, we used both CoCiP[18–20] (a physics-based simulation model) and a machine learning (ML) model trained on a database of automatically detected contrails[21], which attempts to correct for short-comings in the weather forecast data (see Methods section). By using (treatment- and control-group) blinded human evaluators and satellite imagery, we assessed contrail formation on a per-flight basis. We assessed the effectiveness of our forecasts with a crossover randomized controlled trial, where outbound and return flights in a pair were randomized to pass through or avoid the same forecasted CLZs. This design helped us control for confounding factors such as weather conditions, and aircraft engines. Flights in the treatment group adjusted their routes to avoid CLZs, and we observed the resulting flights in satellite imagery.

Simulations of navigational contrail avoidance[6,7] show that small adjustments can prevent persistent contrails. Our methods using satellite imagery do not allow us to test this hypothesis directly, instead we are studying the impact of flight adjustments on the formation of satellite-detectable contrails.

A recent study tested contrail avoidance in the Maastricht Upper Area Control region by adjusting flight altitudes every other day when potential persistent contrails were forecasted[22]. They used satellite imagery and a contrail detection algorithm to assess whether the deviations were successful on average in an ice-supersaturated region (ISSR). While their approach provided insights into the effectiveness of avoidance maneuvers, it required intervention at the airspace level, limiting its applicability for individual airlines to implement specific mitigation strategies.

Our study builds upon and expands these findings by focusing on a targeted, per-flight approach. This allowed us to pinpoint whether detectable contrails formed by specific flights of interest, eliminating the need for full airspace control and enabling airline-level avoidance implementation. Furthermore, our crossover trial design provided robust statistical significance while impacting a substantially smaller number of flights compared to the alternate-day approach used in the Maastricht study[22]. Lastly, our per-flight intervention enabled the airline to track the fuel usage impact of avoidance maneuvers, a crucial factor for operationalizing contrail avoidance strategies.

## Methods
### Flight sample inclusion & exclusion criteria
To select flights for inclusion in the trial, we applied several screening criteria. First, we identified flights whose paths intersected forecasted CLZs. Next, we selected flights that departed a hub (outbound) airport, typically Dallas or Phoenix, to a satellite airport and then returned to the hub (inbound) on the same day. To fly through the same atmospheric conditions on both outbound and inbound flight legs, we restricted the sample to flights where the satellite airport was near a CLZ, and the flight time between hub

and satellite airports was less than 4 hours each way. To enable the PACE software to receive CLZ prediction updates, we also required that participating aircraft have internet connections, these are provided by vendor WiFi networks through a combination of ground-to-air and satellite systems. Note that the PACE software is not directly integrated with the aircraft's systems. The candidates from this screening were selected manually approximately two days in advance. The trial included a total of $n = 44$ flights. Details on these flights are listed in Table 1 of the supplementary material.

This study focused on tactical near-airport detectable contrail avoidance: delaying the plane's climb after takeoff or descending early before landing to avoid flying through the CLZ. Contrail forecasts were not integrated into flight planning systems at the time of the study, so participating pilots were required to avoid contrails en route. Focusing on delaying the climb or descending early was the simplest way for pilots to optimize the route, work with air traffic control and manage other operational constraints. An example of an application of this strategy is in Fig. 1.

We used two methods to forecast CLZs: a physics-based simulation based on the Contrail Cirrus Prediction model (CoCiP)[18,19] implemented in the open-source pycontrails library[23] and a machine learning (ML) system trained on contrail detections and collocated numerical weather data. Both methods use numerical weather forecast data along an advection path as their primary input. CoCiP uses physical processing based on cloud microphysics, while the ML model is a neural network trained using satellite contrail detection labels.

To identify candidate flights for the trial and plan contrail avoidance routes, we generated altitude-specific CLZ predictions for altitudes between 8 and 13 km, at hourly resolution using both the CoCiP and ML models two days before the planned flights. We used European Centre for Medium-Range Weather Forecasts (ECMWF) high-resolution forecast model for weather inputs. Flights were selected based on CLZs for which the two forecasts were in agreement that a CLZ would be near the turnaround airport for both flights. The day of the flight, pilots coordinated with dispatchers and air traffic control to make the recommended vertical flight adjustments based on updated predictions of the ML-based model.

### Contrail likely zone avoidance planning and execution
Prior to the departure of each participating flight, the outbound flight was randomly assigned to either the control group which flew through the CLZ as originally planned, or the treatment group which adjusted its flight to avoid the CLZ. The return flight was assigned to the opposite group to serve as a matched pair. Since flights were chosen with a CLZ near a turn-around, the same plane both flew through and avoided each CLZ 1 to 2 hours apart.

Contrail avoidance flight legs were planned using flight management system software developed by PACE, integrated with the ML contrail forecasts. This system enabled flight planners and pilots to make tactical decisions to avoid contrails, such as manually changing the altitude before takeoff or adjusting the altitude in flight. The platform's interface was modeled on clear air turbulence, a concept already familiar to the pilots.

Figure 1 shows an example of the contrail screen used for an experiment flight in which the pilot delayed ascent after takeoff to avoid detectable

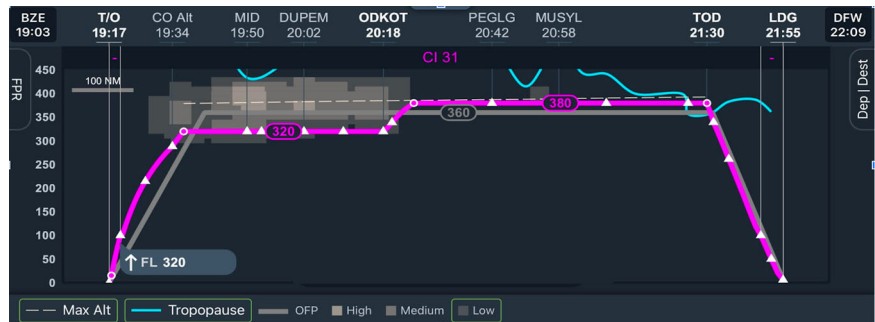

**Fig. 1 | Successful contrail avoidance as seen on the PACE panel.** The PACE panel shows the vertical profile (purple) of a late ascent contrail avoidance maneuver. A contrail likely zone (CLZ) is shown in gray, just above the left side of the flight path. The pilot originally planned to fly at FL360 (36,000 feet), the level of the gray line. By staying at FL320 (32,000 feet) for part of the flight, the CLZ was avoided and no detectable contrails were created.

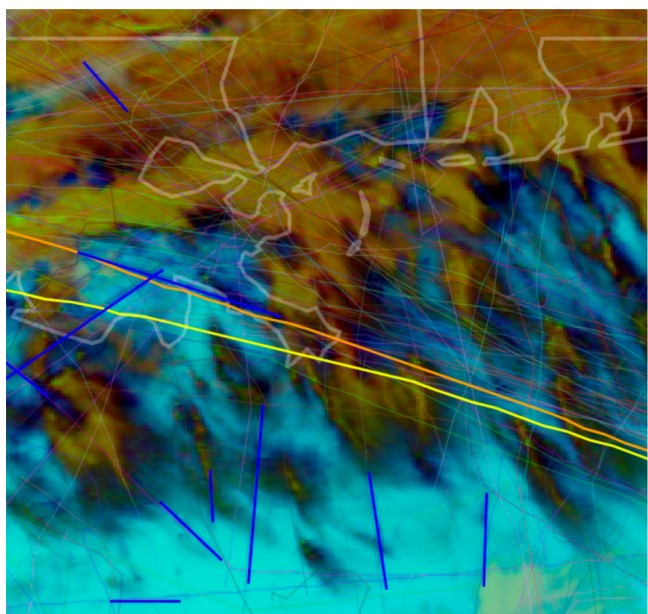

**Fig. 2 | GOES-16 satellite perspective of original and wind-advected flight paths, with contrail detections.** Example of one frame of the GOES-16 satellite imagery sequence over the Gulf Coast area. This was used for labeling whether American Airlines flight 189 created a detectable contrail. Thick lines show the original flight path and wind-advected flight trajectory, along with contrails detected by the computer vision system[21]. Other advected flight paths have a variety of lighter colors on thinner lines. In this case the alignment between the advected flight path and the observed contrail led the evaluators to conclude that this flight made a detectable contrail.

contrails. This flight adjustment decreased the probability of contrail formation because the flight was at a lower altitude during the avoidance treatment than the CLZ lower bound. Note that the aircraft initially climbed to FL320 (pressure = 275hPa, temperature = 235K), which was below the relevant CLZ but still an altitude where contrails can form under certain conditions, and climbed to FL380 (pressure = 205hPa, temperature = 218K) once the CLZ had been passed.

### Satellite image-based verification

To determine whether contrails were created or avoided for each flight, we used a sequence of false-color GOES-16 infrared satellite images, which included wind-advected Automatic Dependent Surveillance Broadcast flight trajectories of both the target flight and other nearby flights at 10-minute intervals. The false-color helped to highlight the presence of clouds, in particular thin cirrus clouds should show up as dark features in the image[21,24]. Access to all image sequences used can be found through links in Note 1 of the supplementary material. Three evaluators (authors of this work) independently assessed whether a contrail was present in each image sequence and whether it was formed by the target flight. The evaluators were blinded with respect to which flights were in the control versus the treatment group, as well as to flight altitude information which correlated with treatment. Evaluators assessed contrail formation at any point along the flight path, not just near the turn-around airports where flight adjustments took place. This was to avoid subjective judgment calls about which portions of the control flights were close enough to the target airports to be labeled.

Figure 2 shows an example satellite image, where the orange line depicts the expected location of the wind-advected flight trajectory over time and the blue lines indicate contrails detected by an automated computer vision system[21]. The linear contrail intersects the wind-advected flight trajectory 30 minutes after the flight passed through the advected airspace. Each labeling task consisted of analyzing sequences of these images, which

were generated for the entire flight path, following each wind-advected flight trajectory for 2 hours post-flight. Evaluators were tasked with judging whether a contrail (which would show up as a dark, linear feature in the image) lines up with the expected location of the flight trajectory. Automated contrail detections were provided to the evaluators, though evaluators had the option to label flights as having formed a contrail if that contrail was visible to them, even if it was not detected by the automated system.

### Randomized crossover trial

We conducted a randomized crossover intervention trial to establish causality between contrail avoidance and contrail presence. In particular, we were interested in testing the one-sided alternative hypothesis that the treatment group (contrail avoidance) would have fewer contrails than the control group (no contrail avoidance). The outbound and return flights of the same turn around airport and CLZ were naturally treated as matched pairs, controlling for confounders such as weather conditions and aircraft engines.

To account for the relatively small sample size, we used nonparametric permutation testing. This approach does not make any assumptions about the distribution of the data. For each matched set of flights, we randomly permuted the treatment and control labels and computed a paired-sample exact sign-test statistic on the binary results[25,26]. We repeated this process 200,000 times to generate the null distribution, and calculated the one-sided p-value as the proportion of permuted datasets in which the test statistic was smaller or equal to the test statistic for the observed dataset. A link to the necessary code and data to reproduce the analysis can be found in Note 2 of the supplementary material.

### Contrail likely zone forecast models

We used two different models to forecast CLZs. The Contrail Cirrus Prediction (CoCiP) model is a physics-based model that simulates contrail formation, evolution and impact using atmospheric conditions, aircraft type, flight path, and other features[18,19]. Based on weather variables provided by a forecast or reanalysis product, CoCiP evaluates whether a contrail will form and persist at a given spacetime location and models its lifetime through the initial downdraft and three-dimensional advection with a second-order Runge-Kutta method. Throughout this process, the model continuously compares with local weather conditions to determine whether the contrail continues to persist or sublimates. The model was configured with Exponential Boost Latitude Correction for humidity scaling, a segment length of one kilometer, an integration time step of 5 minutes, a maximum contrail age of 12 hours, a weather update interval of 1 hour, and a wind shear (dsn_dz) factor of 0.665. After modeling the full evolution of the contrail, climate impact quantities such as radiative forcing are calculated.

We ran a gridded version of the model, which evaluates CoCiP on a regular spatiotemporal grid rather than requiring flight waypoints. In order to translate gridded outputs into CLZs, authors of this work examined visualizations of the energy forcing of each grid point predicted by the model[20] and assessed its agreement with the ML model prediction. We used ECMWF's high-resolution forecasts as input to CoCiP[27]. The CoCiP model also uses cloud microphysics to determine which contrails persist, accounting for initial downdraft, fall, and sublimation.

A drawback of using weather forecast data to predict contrail formation is that existing weather products are often incorrect about the locations of high-altitude ISSRs[14,15], with previous works finding that more than 80% of high-altitude ISSRs predicted in ECMWF ERA5 are found by in-situ measurements to be incorrect[13]. It has been suggested that ISSR prediction skill could be improved by using other atmospheric variables as dynamical proxies, which have been shown to be correlated with ice supersaturation[28]. In this work we use these dynamical proxies to improve a prediction model by training a neural network to predict contrail formation. For a given flight waypoint, the neural network takes as inputs not only the weather quantities directly related to contrail formation (humidity, and temperature) but other weather variables: wind velocity, relative vorticity, fraction of cloud cover, cloud ice water content, specific snow water content, and divergence. We

also used local solar time, day of year, latitude, longitude, and altitude of flight waypoints as input features.

To train, validate and test the model, we use a dataset produced to study contrail formation on a per flight basis[15]. This dataset comprised all flights available in ground-based Automatic Dependent Surveillance Broadcast data from FlightAware[29] over a region roughly spanning the contiguous United States from 28 different randomly selected days in the time period Apr 4 2019 - Apr 2 2020. To avoid train-test contamination, the flights in each set were separated by at least one day as in the original study[15].

In particular, this dataset is created by advecting the flight trajectories using ECMWF high-resolution numerical weather forecast wind data and a third-order Runge-Kutta method[30], and then comparing these advected trajectories to automatically detected contrails[21] from the GOES-16 Advanced Baseline Imager infrared images. The result is a set of ≈ 6 million flight segments, each of which is labeled as matching or not matching a contrail. The neural network model used the individual flight segment data and their associated weather fields noted above as training examples. It was trained by minimizing the cross entropy loss of its predicted CLZ probability against the binary label of whether it matched a contrail or not, using stochastic gradient descent. When evaluated on the contrail formation on a per flight basis dataset[15] the model's precision (for a given recall) was about twice that of the weather forecast models evaluated, i.e. when the ML model predicts a flight will match a contrail, that prediction is about twice as likely to agree with the dataset as the physics-based models evaluated in this same contrail formation on a per flight basis study[15].

The ML model was a four hidden-layer fully connected classification neural network with leaky rectified linear unit activation functions and a sigmoid function in the final layer. Dropout was used for regularization. After training, the model's predictions were used as a proxy of GOES-16-detected contrail formation likelihood. These scores were available at every point of a latitude-longitude-altitude voxel grid given numerical weather prediction features along the gridpoints' advection paths.

In order to better understand the ML model, we performed an ablation study by removing weather features from the model and measuring it's performance, computing the area under the Receiver Operating Characteristic curve). These scores vary between 0.5 (random predictions) and 1 (perfect predictions). We expect that removing more important features will lower the score more than removing less important ones. The results can be seen in Table 1. The analysis agrees with earlier studies: we find that humidity data is the most important feature. Second most important is cloud data, which is closely related to humidity, but also the presence of clouds may affect whether we can detect a contrail. Vertical transport is third-most important, such quantities have earlier been suggested as dynamical proxies for contrail formation[28]. Note that correlations between weather quantities may allow the model to estimate quantities even if they have been nominally removed. This may explain why, for example, removing humidity data still allows the model some success in predicting contrails even though humidity is a crucially important quantity. Note that removing both humidity and cloud formation information, which is highly correlated with it, leads to a much larger performance drop.

Figure 3 shows an example of forecasts from the ML model (left) and CoCiP model (right) used to select candidate flights. When choosing flights to participate in the experiment, both the ML and CoCiP models needed to show a CLZ on the flight path. In this example Chicago ORD was a good candidate for a turnaround airport as shown by the CLZs in both forecasting systems. Once a flight has been selected, when adjusting the flight to avoid CLZs (as in Fig. 1).

### Post-Flight Verification

To determine whether contrails were created or avoided for each flight in the trial, three evaluators (authors of this work) independently completed post-flight analyses to assign individual binary labels. Figure 2 shows an example of the satellite imagery used. The yellow line shows the flight path, which was obtained from Automatic Dependent Surveillance Broadcast data licensed from FlightAware[29].

The evaluators assessed whether a flight made a contrail based on the following criteria:

- Proximity and direction: How close is the contrail to the advected flight trajectory, is it aligned in the same direction?

**Table 1 | Ablation study of the contrail prediction ML model**

| Features removed | Area Under the Receiver Operating Characteristic Curve | Δ |
|---|---|---|
| None | 0.855 | |
| Humidity (q,r) | 0.841 | 0.014 |
| Cloud data (ciwc, cswc, cc) | 0.846 | 0.009 |
| Vertical transport (d,v,w) | 0.847 | 0.008 |
| Temperature (t) | 0.853 | 0.002 |
| Horizontal transport(u,v) | 0.854 | 0.001 |
| Humidity and clouds (q, r, ciwc, cswc, cc) | 0.792 | 0.063 |

The symbols inside the brackets are the names of the quantites in the European Centre for Medium-Range Weather Forecasts Integrated Forecasting System data. The first row shows results using all the features we use, other rows show the effects of removing some features.

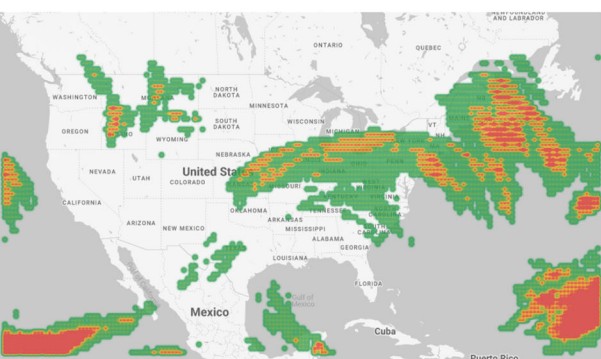
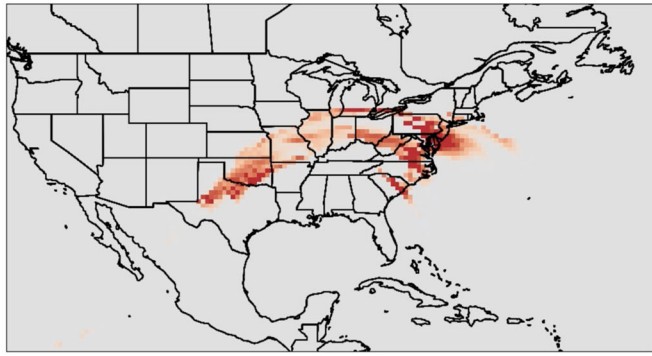

**Fig. 3 | Machine learning and CoCiP based contrail likely zone forecasts.** Example contrail likely zone forecasts (CLZ) used to select flights for Thursday, March 23, 21:00 UTC from the ML model (left) and the CoCiP model (right). The left image shows all-altitude (FL 260-FL420), color-coded CLZ probability thresholds, with red, yellow, and green corresponding to high, medium, and low probabilities of contrail formation, respectively. The right image shows the forecast for flight level 360, with blue coloring indicating net-cooling contrails and red coloring indicating net-warming contrails. The contrail's net impact was not considered for the purposes of this trial.

- Presence: Is there a high level of confidence that the object in the frame is a contrail?
- Timing: Did the suspected contrail first appear 20-40 minutes after the flight passed?
- Persistence: Is the contrail visible in multiple frames?
- Speed: Does the suspected contrail move at the same speed as the advected plume of the flight?
- Exclusivity: Are there no other flights in the FlightAware database that are a substantially better match for the contrail in question?

We note that the criteria were used as guidance, as there were cases for which it was not possible to assess all criteria and others where evaluators determined matches even when one or more criteria were not met. The timing criterion was present since it often takes some time for contrails to become large enough to be detected in at least two consecutive GOES-16 frames. Like all the criteria this is guidance only, as contrails can appear much earlier than 20 minutes under certain weather conditions and temporal alignments with satellite scans[31].

To account for uncertainty in contrail avoidance compliance, the evaluators assigned a binary label based on whether a contrail was created anywhere along the flight path above 415 hPa (6915 meters), typical contrail formation altitudes[32]. Disagreements among evaluators were resolved by majority vote. See Fig. 4 for a geographic coverage of the flight sample.

## Results

### Ascent/descent adjustments lead to a decrease in detectable contrail formation

As shown in Table 2, the treatment group, which aimed to avoid contrails, made 4 detectable contrails, 63.6% fewer than the 11 observed in the control group. We rejected the null hypothesis of no change in detectable contrail formation between the treatment and control groups (p = 0.0331, permutation rank sign test). These results suggest that the early descent or delayed ascent informed by the CLZ forecasting system caused a statistically significant reduction in detectable contrail formation. They also demonstrate the operational feasibility of detectable contrail avoidance on a per-flight basis.

Detectable contrails accounted for 1.97% and 0.89% of total flight kilometers in the control and treatment groups respectively. This represents a 54.4% reduction in detectable contrails per flight kilometer in the contrail avoidance flights. We tracked fuel usage throughout the experiment using aircraft movements and inflight position reports sent through Aircraft

Communication Addressing and Report System via Very High Frequency and Satellite Communications. The treatment group used an average of 2% more fuel per adjusted flight, corresponding to an additional 0.26 kg of $CO_2$ emissions per kilometer of treatment group flight.

## Discussion

Our work has shown a reduction in observed contrail formation with a treatment group size of 22 flights. We opted for a tactical near-airport contrail avoidance intervention for several key reasons: 1) This approach allowed for an effective crossover trial design, as we had flexibility in selecting destination airports near CLZs. This minimized the time difference between outbound and inbound flights encountering the same CLZ, thus controlling for potential confounders like aircraft type and atmospheric conditions. 2) Near-airport maneuvers simplified operations and ensured higher pilot compliance. 3) Compared to modifying cruise levels, which could require prolonged low-altitude flight or multiple altitude changes, delayed ascent/early descent minimized additional fuel usage by avoiding extra climbing. Additionally, lower altitudes are associated with a reduced probability of contrail formation even in the presence of some ice supersaturation. It is worth noting that, although statistical significance is influenced by sample size, it is also heavily dependent on the signal-to-noise ratio of the data[33]. Our crossover design trial together with the tactical near-airport avoidance intervention mitigated the change in atmospheric conditions present in treatment and control groups, thereby controlling for potential weather-related and other confounders which helped reduced noise[34]. Given the small sample size, we avoided asymptotic assumptions about normality and adopted a non-parametric approach, enhancing the statistical power of our hypothesis test in this context[25,26].

An important future direction would be to show this method extends to larger scale, such as by performing a similar experiment with hundreds or thousands of flights. In addition to demonstrating operational feasibility, such a trial could expand the route selection criteria to allow testing avoidance of mid-flight CLZs, horizontal as well as vertical path adjustments, geographic regions with different weather patterns (e.g. North Atlantic or subtropical routes) and nighttime flights which can have a particularly strong warming impact. Moreover, future trials might consider relaxing the requirement for agreement between ML and CoCiP models in CLZ identification to explore contrail avoidance feasibility under a broader range of prediction scenarios. Eventually, trials should consider the radiative forcing impact of contrails when making flight-adjustment decisions, so that the focus can be on evaluating the feasibility of minimizing the warming impact as opposed to just contrail formation. In this experiment, we validated detectable contrail avoidance manually for each flight, but a larger trial would likely need to use an automated system[15,31].

There are a few important factors to consider when interpreting the results. For example, having observed 11 contrails in the control group, where 22 flights flew through CLZs, suggests CLZ forecasting is challenging and can lead to false positive CLZs. Weather forecasts of humidity at contrail-relevant altitudes are subject to inaccuracies[13,16,35]. We control some of this uncertainty by using two different approaches for CLZ forecasting (physics based and empirical-based ML), and with our inclusion/exclusion criteria for the flight sample. The 50% rate of contrail observation for flights predicted to form a contrail in the control group is relatively high compared to previous approaches[13-15]. It is also worth noting, that the probability of the ISSR not being present on the turnaround leg is balanced between the treatment and control groups due to the randomization of the crossover trial

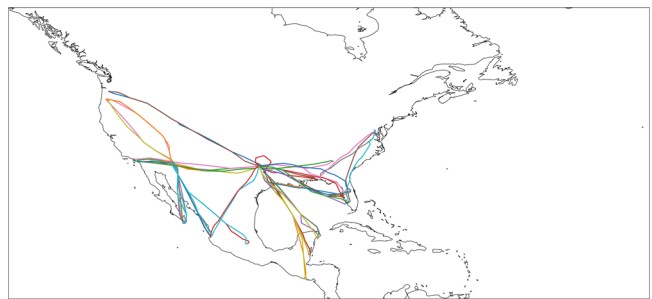

**Fig. 4 | Flight sample considered in the randomized crossover trial.** Geographic coverage of flights considered for the trial, colors are used to illustrate different flights.

**Table 2 | Number of flights with GOES-16 detected and undetected contrails, total detected contrail length, and total flight distance for the control and treatment groups**

|  | No detectable contrail created | Detectable contrail created | Sum of contrail km | Sum of total flight km |
|---|---|---|---|---|
| Control | 11 | 11 | 726 | 36802 |
| Treatment | 18 | 4 | 321 | 35729 |

design. Therefore, it should not systematically bias the overall conclusions of the trial.

Difficulties in CLZ forecasting may also explain the observed contrails in the treatment group. Alternatively, these false positives could have been caused by attributing the contrail to the wrong flight, which can happen in image sequences with a high density of flight paths and contrails such as the example in Fig. 2. False positives could also be attributed to our labeling strategy; our evaluation was conservative in that it did not take into account where in the flight the contrail was made: a contrail formed in any part of the flight path was counted towards the treatment group, even if formed after a successful CLZ avoidance intervention to ensure objective evaluation as mentioned in the Results section. Some contrails may not be visible in the images, either because of the chosen infrared channel color scheme, because the 2 km satellite resolution might not show faint or optically thin contrails or they might have been simply blocked by higher clouds[15,21]. Finally, some flights (e.g. military flights) may be missing from the Automatic Dependent Surveillance Broadcast flight trajectory data provided by FlightAware[29], which could complicate flight attribution.

Previous simulation studies[6,7] have argued that small-scale flight deviations can avoid creating contrails, but this work's use of satellite imagery for evaluation tested whether satellite-detectable contrails were avoided. Research on the radiative properties of contrails could help to quantify under what conditions contrails form but are not detectable by satellite.

## Conclusion

In this work we performed a randomized control trial of whether small-scale flight deviations can reduce detectable contrail formation. Using machine learning and physics-based CLZ forecasting models, we found a statistically significant reduction in the number of observed contrails in the flights that attempted to avoid contrail forming regions. This study provides a proof-of-concept that commercial airlines can verifiably avoid detectable contrail formation, one of the first steps towards developing a comprehensive avoidance strategy. We hope that these findings motivate further research into contrail avoidance via flight route planning at a global scale.

## Data availability

The supplementary material contains a table with flight-level information, and publicly available links to all flightpath visualizations used by evaluators. CoCiP simulations can be reproduced using the pycontrails API at api.contrails.org.

## Code availability

See Note 2 on the supplementary material for the link to the publicly available Google Colab notebook used for the statistical analysis carried out with Python 3.11.8 using the following libraries: NumPy[36], Pandas[37], SciPy[38], tqdm[39], seaborn[40], Matplotlib[41].

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

## Author contributions

The study scope was defined by C.E. and C.V.A., A.S.W., C.E. and S.C. developed the randomized trial design, while C.E. and J.D. designed the trajectories. Weather data for CLZ forecasting was acquired and processed by C.E., U.H., and M.S. The CLZ CoCiP prediction model was developed by M.S., T.D. and Z.E. The ML prediction model was a collaborative effort by C.E., S.G., T.S., A.S., N.G., U.H., and J.N. C.E., A.S.W., S.G., A.S., K.M., and N.G. developed the post-flight verification protocol for contrail presence. C.E., F.O. and M.J. designed the UX for PACE/Google/AA integration. Pilot selection and training were handled by C.E., T.D., and D.S., J.D., D.H., M.G., J.B. and R.G. C.E., O.S. and N.A. designed and developed the user interface for displaying ML prediction forecasts. C.E., S.G., A.S., C.V.A., K.M, and E.B. acquired and pre-processed the satellite imagery and flight trajectory data, while C.E. and C.V.A. and M.G. acquired non-trajectory information for specific flights. C.E., S.G., A.S., K.M., C.V.A., N.G., and A.S.W. Analyzed the satellite imagery to assess contrail formation in each flight, and C.E., J.D., D.H., M.G., B.N. and B.B. determined the specific trajectories to be flown. A.S.W. conducted the statistical analysis. C.E., A.S. and C.V.A. built tools for visualizing contrail formation. A.S.W. interpreted the results and wrote the manuscript. C.E., A.S.W., S.G., A.S., K.M., N.G., M.S., T.D., E.B., and D.S. refined and edited it. C.E. and A.S.W. created the figures and tables summarizing the results. Finally, C.E., J.C.P., C.V.A., N.G., E.B., D.S., J.R., R.S, J.B, and R.G. managed the project administration. All authors reviewed the manuscript.

## Competing interests

The authors declare the following competing interests: As denoted by their affiliations, some authors are employed by Google Inc., Breakthrough Energy, LLC, PACE Aerospace Engineering and Information Technology GmbH, and American Airlines. All other authors declare no competing interests.
