## [Transparent Peer Review file · Communications Engineering]

Feasibility Test of Per-Flight Contrail Avoidance in Commercial Aviation

Corresponding Author: Dr Carl Elkin

Version 0:

Reviewer comments:

Reviewer #1

(Remarks to the Author)

The paper deals with a very interesting topic which attracts high attention currently. The paper is written well and the overall aims of the work performed are reasonably described.

However, I recommend major revisions of the manuscript comprising in particular the conclusion section, because they are critical for the scientific content and technical soundness.

1. Is the paper technically sound?

Overall concept of alternative flight trajectory planning is a well-established method in the existing literature. Using satellite imagery for contrail detection is also well established. Similarly, combination of both for evaluating success of contrail avoidance strategies during real flights is also a well-established methodology and has been explored, implemented and demonstrated in other studies, e.g. Sausen et al., 2023 and Hofer et al., 2024. Overall, the paper is technically sound in describing its overall approach, however we recommend further clarity in terms of the statistics issue with such a small sample of flights – only 22 flights each group, in the light of the much to generalized conclusions drawn.

2. Are the claims convincing? If not, what further evidence is needed?

The work presents a workflow how aircraft could avoid CLZ regions relying on information as forecasted by models. It is recommended that authors stick to this clear description, and hence avoid to state that they are avoiding contrails (e.g. line 142), and avoid to describe displays while stating that no contrails were created (i.e. line 109), instead of indicating that they simply those regions where CLZ were forecasted were avoided. From the description I get the impression, that no validation or other confirmation of the real conditions, e.g. in terms of “skill of the forecast” or similar was performed. Hence, the authors of the study did not explore to what extent they were not able to observe an contrail, because in the real atmosphere no CLZ region existed. They count the non-existence as a success

3. Are the claims fully supported by the experimental data?

As result from my review, I come to the end, that the claims are not supported, critical points are not adequately assessed. By way of example, e.g. no methodology has been proposed in order to identify false forecast information, which could lead to wrong conclusions. To illustrate, the fact that no contrail is formed, could also be due to a wrong forecast of CLZ; and not as concluded by the authors, on a successful avoidance strategy. Additionally, the fail to mention that their estimates when quantifying climate effect of an avoided contrail relies only on parameterized numerical simulations.

Furthermore, I suggest major revision of the claims, more specifically:

Line 164f: The authors conclude that they have delivered a proof of concept for an avoidance of contrails and that they have shown that contrail avoidance is a viable strategy to combat climate change. To my understanding this is not supported.

Line 168f: Further, they claim that this study is one of the first steps towards developing a comprehensive avoidance strategy. In the light of previous literature, this is seen to be kind of misleading. It is probably more adequate to consider this study as one further step towards developing a comprehensive avoidance strategy, more specifically they have completed an initial experiment on tactical contrail avoidance.

4. Is the statistical analysis of the data sound?

Line 234f: The authors are presenting results from quite a small sample size, i.e. 22 flights in each group. However, I have the impression that they fall short on proposing an adequate methodology, in order to proof the statistical significance of their results, which they have achieved with such a small sample.

5. Does the availability of data adhere to the expected standards of your research community?

Access to data

Line 321 In the manuscript the explanation “The datasets used and/or analyzed during the current study are available from the corresponding offer on reasonable request.”, to my understanding does not make sense. It should be probably noted author here, instead of offer.

However, it is strongly suggested to upload the data to a repository, e.g. zenodo, or similar, assuring reproducibility of this study.

6. Are the claims appropriately discussed in the context of the previous literature?

Line 42: Early literature is not taken into account, e.g. Hermann Mannstein, Peter Spichtinger, Klaus Gierens, A note on how to avoid contrail cirrus, Transportation Research Part D: Transport and Environment, Volume 10, Issue 5, 2005, Pages 421-426, ISSN 1361-9209, <https://doi.org/10.1016/j.trd.2005.04.012>.

(<https://www.sciencedirect.com/science/article/pii/S136192090500026X>), but also Green et al. 2003 is not mentioned.

Line 28: Using the Lee et al. 2021 is not adequate for providing reference on an “extremely cost-effective way to reduce anthropogenic climate forcing”, hence this reference needs to be deleted here.

7. If the manuscript is unacceptable in its present form, does the study seem sufficiently promising that the authors should be encouraged to consider a resubmission in the future?

Major revisions are suggested.

8. Is the manuscript clearly written? If not, how could it be made more accessible?

Yes, the manuscript is clearly written.

9. Are there any special ethical concerns arising from the use of animals or human subjects?

Not applicable.

Reviewer #2

(Remarks to the Author)

The authors report on a study that tests the feasibility of contrail avoidance by flight rerouting, a topic that has gained much attention lately. The question being tested is if ice supersaturated regions can be forecast with enough precision to be able to reroute flights in order to avoid contrail formation. The authors study 11 ice supersaturated areas, for which the forecast of a ML algorithm and a physically based model agree, reroute flights crossing those ice supersaturated areas and use satellite observations to study whether the flight rerouting was successful. I appreciate the effort that went into this work but the study makes a number of assumptions that limit the significance of their work.

The authors are careful to write in the main text that they check for ‘detectable’ contrails, in large parts of the paper the word ‘detectable’ is missing and instead the authors write about ‘contrail avoidance’ instead of ‘avoidance of detectable contrails’. This is not only a wording but a science issue. If the authors write that they avoid contrails then they suggest that the whole climate impact connected with the contrail has been avoided. If, on the other hand, it is simply the contrail’s optical depth that has been reduced to below the detection threshold then only a part of the contrail climate impact has been avoided. Since contrails are typically optically very thin clouds, the difference in terms of optical depth may not even be large. Furthermore, it may be that the contrail’s ice water content and with it the optical depth increases more slowly after rerouting so that they become visible after the maximum 40 min., the maximum time delay allowed in this study. Differences in contrail optical depth and its development may be connected with differences in the atmospheric conditions at the lower flight level relative to the higher flight levels that control contrail ice nucleation and water vapor deposition. Therefore, by flying lower it may either be that contrails do not form or that contrails form but with changed optical depth which may make them not detectable or detectable only later in their life cycle. In any case, ‘contrail avoidance’ cannot be shown, at least not with the methods used here.

The motivation of this work is the assumption that contrail avoidance is a promising climate change mitigation strategy. This has never been shown conclusively. It is well known that the simulation of clouds is the largest obstacle in climate modelling and weather forecasting. The uncertainty connected with cloud properties and relative humidity is large and translates into a large uncertainty in contrail properties and their climate impact. The present paper shows that the avoidance of detectable contrails can be achieved by a ‘mere’ 2% increase in fuel per adjusted flight. This additionally emitted CO₂ will have an impact on radiation for centuries and the gain will be the avoidance of a short-lived climate impact of uncertain size that is connected with the avoidance of detectable contrails. But of course, in some (or many?) cases, rerouting does not have the desired outcome and the 2% increase in CO₂ emissions may be in vain. Increasing the climate impact of long-lived climate forcers in order to reduce the impact of short-lived climate forcers of uncertain impact is potentially very dangerous. The fact that not all climate change components connected with aviation, that may also change due to rerouting, have been quantified complicates the problem further. This means that the present paper studies the feasibility of a mitigation option for which the resulting change in the climate impact of aviation is currently impossible to robustly quantify.

Figure 2 shows that from a large number of flights only very few form detectable contrails even though the length of the detectable contrails formed would indicate a larger ISSR. Even in the vicinity of those detected contrails other flights appear not to form a detectable contrail. Detectability of contrails depends on many variables. A larger fuel use leads to an increased contrail optical depth which makes it easier to detect the contrail. A higher altitude leads to a larger temperature difference between surface and contrail which impacts the signal in OLR. Many of the detected contrails end in the area with natural cloudiness over an island. Lower cloudiness and changing surface conditions are known to make contrail detection more difficult. It is very likely that many more contrails formed but were or could not be detected. The failure to detect a contrail does not mean that the contrail is not there or has no climate impact. This supports the earlier argument that the mere fact that a contrail cannot be detected does not mean that it has been avoided.

There is next to no model description for CoCip other than it is a ‘physics based’ model. Please expand.

How are CLZs calculated within CoCip?

Your neural network approach uses many input variables. Which variables are most important for the successful prediction of contrail formation? Which variables are not important? Does your analysis agree with earlier studies?

By choosing ice supersaturated areas for which two very different forecasts agree, the authors are picking situations for which the predictability is high and the ice supersaturated areas are likely very large. This means that the authors are not randomly sampling but choosing systems for which their approach is most likely to work. Therefore, the results of this study should not be interpreted as proof of concept after which only upscaling is required.

You write that 'ISSRs are rare and cover a low portion of the upper troposphere. Therefore, only a small percentage of flights would need to make small adjustments to avoid the majority of contrail warming': please be more precise and give numbers and discuss how certain those statements are.

What are the predicted spatial and temporal scales of the ISSRs you are trying to avoid?

Contrails absorb and reemit outgoing longwave radiation.

Line 91: please give pressure and temperature at flight level and not just FL320.

Line 143-144: the true positive could also be due to attributing the contrail to a wrong flight.

Lines 164-168 are simply claims not supported by any evidence within this paper. If the authors want to claim that the contrail warming is larger than the warming due to the additionally emitted CO₂ then they should include an estimate of the contrail warming and its uncertainty and estimate the overall climate impact of the flight with/without rerouting. Furthermore, they should include a description of the method how they are calculating all climate impact components associated with the flight and discuss the uncertainties.

Reviewer #3

(Remarks to the Author)

The paper examines the key challenge of the feasibility of contrail avoidance in commercial aviation. The published results are an important contribution to the topic and show that efforts in this research area should be continued to confirm the positive outcome on a larger scale and to include more elements of the air traffic control network.

The main claim of the paper is that it is operationally feasible for commercial flights to avoid contrail formation on a per-flight basis by making altitude adjustments based on predicted contrail-likely zones (CLZs). This claim is based on the use of prediction tools developed by the authors in previous research, namely a combination of a physics-based simulation model and a machine learning model to perform the prediction task. These predictions guide pilots and air traffic control in making altitude adjustments to avoid these zones in both the strategic and tactical phases. To focus on the feasibility of the avoidance strategies on a flight-by-flight basis, the authors chose to focus on conditions where full air traffic control was not required, thus simplifying both the operational method and the statistical analysis.

The test results were performed by a rigorous randomised controlled trial (RCT) comparing the frequency of occurrence of contrails for a set of flights diverted to avoid CLZs (the treatment group) with the frequency of a control group. The main difficulty with the statistical analysis comes from the presence of confounding factors (weather conditions), the small sample size (22 flight pairs) and the fundamentally counterfactual nature of the null hypothesis being tested. To mitigate this difficulty, the authors decided, in a very pragmatic way, to focus on pairs of very close outbound and return flights, to assign the outbound flight to a group and the return flight to the opposite group, thus trying to reduce the role of confounding factors.

From an experimental design point of view, we believe that the 'matching' method used here is particularly clever and effective in reducing the role of confounding factors, especially given the small sample size (22 pairs of control/treatment flights). The risk obviously lies in the change in weather conditions during the 2 hours between the outward and return flights. In future work with significantly more flights, it might be interesting to compare other approaches to mitigate these effects (stratification, regression-based statistical adjustments...).

From a statistical point of view, the use of non-parametric tests is fully justified here given the small sample size (Bayesian approaches were also possible), increasing the test statistical power and confidence in the final p-value. However, again, a drastic increase in sample size in future work would be the only way to really confirm the results.

Although the experimental design is perfectly described in the paper, we would like to clarify our understanding of the sample construction. We understand that the selected flights (whose trajectory crossed a CLZ) were selected based on a prediction two days before the flight, but that the avoidance strategy was adjusted on the day of the flight. Does this mean that the 22 CLZs predicted two days before were confirmed by a new prediction two days later (which seems to contradict slightly the 50% precision in the control group), or are the 22 flights the result of a subsequent filter? However, this question is not crucial as the article does not formally address the issue of the prediction horizon.

In conclusion, this paper is an important contribution to the field, with plausible and positive results that need to be confirmed in subsequent research with more flights and a more complete air traffic control network. The statistical methodology and experimental design are particularly well suited to the difficult task at hand (statistical hypothesis testing with confounding factors and a small sample size).

Reviewer #4

(Remarks to the Author)

Thank you for the opportunity to review this excellent paper. The design of experiment, the methods/tools used, and the results are all contributions to the literature and advancing this important work.

Please accept the following minor comments;

(1) Abstract/Intro; refers to "major" "significant" and "substantial" contribution of contrails to total anthropogenic impact. This seems to be overstating the case given that estimate of contrails contribution is 2% (Lee et.al. 2021). Why not just say 2%

(2) Intro, para 3 "often struggle" Is this a scientific term? Perhaps "subject to inaccuracies" and reference Gierens, Geraedts, handen, Reutter,...

(3) Intro, para 3 "navigational contrail avoidance" term is used here. This is a key concept. What is it?

(4) Intro, Para 3. This para discusses the challenges of implementing Contrail Avoidance (i.e. predicting ISSR and airline systems). Between these two is mentioned of the difficulty designing an experiment (line 33, 34). This is confusing.

(5) Intro, Para 4. Discusses Saussen "Complete air traffic control" What do the authors mean by this. As a pilot in controlled airspace, the flight is always under complete air traffic control. Perhaps what is being referred to is that ATC determined the CLZ and directed the flights away from them. i.e. this was all done by ATC (not the airline).

(6) Intro Section. There are two paragraphs buried in the Discussion and Conclusion section paras 4 & 5, that do nice job in explaining the motivation for this study. It extends/complements the Saussen et al study by using only airline ops independent of ATC. These 2 paras are well written.

(7) Intro identifies a "physics-based model" Why not identify this a CoCIP and credit Schumman. Also explain that the (atmospheric) physics-based model uses atmospheric forecast data on the flight-path to predict whether Contrail will occur.

(8) Intro identifies a "machine learning" model. This is very vague. Perhaps explain the ML model supplements the CoCIP atmospheric physics-based model. See description in Methods section para 2 and 3.

(9) Intro, last para, 2nd sentence. Confound the purpose of the study (i.e. per flight basis) with the detail of the design of experiment (i.e. blind human raters).

(10) Results, para 1. "CLZ predictions over the internet" ??? Somewhere else in the paper is mentions ACARS and SATCOM. This is important, cause ACARS and SATCOM have limited bandwidth and are expensive.

(11) Results, para 2. The discussion on "tactical near airport contrail avoidance" is VERY important. This is opposed to Cruise tactical flight level changes. Why was this approach used. This reviewer assumes that it was applied since it avoided amending the Cruise Flight Level that would require interaction with dispatch and air traffic control. Is this hypothesis correct.

This approach requires flying at lower than planned altitudes, resulting in additional fuel burn. A cruise flight level change may reduce the additional fuel burn. This should be explained.

(12) Results, para 4, Add nautical miles 8km (4.3 nm) and 13 km (7 nm). These are very short contrails? No?

(13) Results, para 4, Is there any data that can be reported on their duration (e.g. 2 hours).

(14) Results, para 4. How long do ISSR last? Is two hours short enough that the ISSR will still be there?

(15) Results, CLZ Avoidance Planning and Execution, para 2 is this "clear air turbulence" or just turbulence

(16) Results, Sat-Image Verification, para 2. How does the satellite image show the "exhaust plume" What is the satellite image actually showing.

(17) Results, Sat-Image Verification, para 2. Last few sentences are confusing. What are authors trying to say.

(18) Results, Ascent/Descent Adjustments, para 2, 1st sentence. Confusing. Why is this metric useful. Contrails contribute to <2% of the stage-length. The treatment was approx 1%. So the treatment reduce the contrail length by 50%. Is this correct?

(19) Results, Ascent/Descent Adjustments, para 2, last sentence. How was this estimate derived. Please show equations/calculations. Also 106 kg?

(20) This reviewer does not understand why the Methods section appears after the Discussion and Conclusions section. This information would have been useful in the Results Section.

(21) Also, the Results section is really a description on of the Methodology (i.e. Design of Experiment). The design of Experiment is unique contribution to the literature and should be described clearly, such that future papers can refer to this.

(22) What is the Conclusions of the study? What are the limitations of this study? What is future work? State these clearly

independent of the Discussion.

Version 1:

Reviewer comments:

Reviewer #2

(Remarks to the Author)

The paper studies the feasibility of contrail avoidance. It concentrates on operational issues avoiding CLZs (contrail likely zones) and on the associated changes in satellite-based contrail detection. I do not have a problem with the main work and agree with the authors that the size of the data set is rather limited. Additionally, the paper makes some unsupported claims about the climate impact of this mitigation option that need to be removed.

The authors say in line 43-45 'We focused exclusively on demonstrating the feasibility of contrail avoidance on a per-flight basis and did not consider the radiative forcing of avoided contrails'. This appears to be a fair description of their work and, therefore, any claims that 'the work suggests that contrail avoidance could lead to a significant reduction in the climate impact of aviation' (last sentence of the abstract) are unsupported and should be cut.

The authors frame their work by mentioning only 'uncertainty surrounding operational constraints and accurate formation predictions' (2nd sentence abstract) as problems connected with proposing contrail avoidance as a mitigation option. This subject is expanded on in the paragraph starting in line 33. In line 174-175 it is simply stated that 'A larger trial with more statistical power could also quantify the amount of warming prevented through contrail avoidance, allowing estimates of the cost-effectiveness of contrail avoidance ...'. This seems to imply that it is straightforward to estimate the climate impact of contrail cirrus and that estimates do not suffer from uncertainties. The conclusions finish by saying that the authors 'hope that their study motivates airlines and policy makers to avoid contrails'. Even though I understand that the climate impact of 'contrail avoidance' is not the subject of this paper, it is necessary to relate the topic of this paper to the bigger picture. I have seen no discussion about the problems and uncertainty when estimating the climate impact of contrails and contrail avoidance, when comparing long-lived and short-lived forcers..... In the contrary, the authors appear to suggest that the 'quantification' of the climate impact is straightforward. This is despite the fact that some relevant papers are cited. The two Lee et al papers describe a large number of uncertainties and can lead the authors to papers describing uncertainties in more detail. Finally, the authors failed to acknowledge the uncertainty coming from the fact that contrails may not have been avoided but that their optical depth may be decreased so that they are not detectable any longer while still having a climate impact (see my last review).

Overall, the paper strengthens a growing body of literature of the feasibility of contrail avoidance. I don't see the paper as an important step forwards since the main road block is how to quantify the climate impact of the mitigation option.

Additional major comments:

Line 27-28: From the fact that ISS can be found in 13.5% of cases in the upper troposphere and 2% in the stratosphere you cannot conclude that between 2% and 13,5% of flights form a contrail. After all, during a flight, contrails will form only in a small fraction of the flight distance but the number of affected flights can be large. Furthermore, all flights that are in the stratosphere need to have passed through the troposphere so that a 2% probability of contrail formation is significantly too low. Additionally, a newer paper from Petzold et al 2020 looks at in-situ IAGOS data and finds that over all of the analyzed areas (Europe, North Atlantic and eastern North America) 'air masses close to the tropopause level are nearly saturated with respect to ice and contain a significant fraction of ISSRs with a distinct seasonal cycle of minimum values in summer (30 % over the ocean, 20 %-25 % over land) and maximum values in late winter (35 %-40 % over both land and ocean)'. The authors need to update ISS frequencies and need to acknowledge that more flights need to be diverted in order to reduce the contrail climate impact.

Line 53: The following statement is wrong 'Simulations of navigational contrail avoidance find that avoiding ISSR regions prevent contrails from forming'. It is well known that ISS is not needed for contrail formation – it is only needed for contrail persistence. Furthermore, knowledge of contrail formation conditions is not a result of simulations conducted in connection with contrail avoidance. This is based on our understanding of cloud physics and supported by observations. There are a number of papers by Schumann et al. on this topic, e.g. the Schumann 2005 which you cite already.

Line 143: You say that in around 1-2% of total flight km a contrail was formed. This value is surprisingly low even if a large part of the flight km were flown within the stratosphere - which appears not to be the case in your figure 1. Since temperatures in most of the northern hemispheric main air traffic areas are such that contrail formation conditions are met when air is ice supersaturated, this means that the formation probability of detectable and non-detectable contrails is much larger than what you estimate. Apparently, you are only observing the contrails with the largest optical depth and missing the bulk of contrails which appear to remain non-detectable (see Kärcher et al. 2009 'Factors controlling contrail cirrus optical depth'). It is important to acknowledge the existence of a large number of non-detectable contrails as they explain part of the climate impact of contrail cirrus.

Line 148-149: In the supplementary material you give details about your calculation of the contrail cirrus equivalent CO₂ emissions. You divide contrail cirrus ERF by the AGWP of CO₂. This calculation does not give the equivalent CO₂ emissions connected with contrail cirrus. You should have divided the AGWP of contrail cirrus with that one of CO₂ in order to get equivalent CO₂ emissions. Please note that in several places in the supplementary material units are incorrect or

missing.

Line 154-155 and 161-162: by targeting the same CLZ with outbound and inbound flights you may not eliminate the problems connected with changing atmospheric conditions. Moisture fields are very small scale and can significantly vary on small spatial and short temporal scales.

The model description does not yet cover all the necessary information, e.g. it is important to say that Cocip is a plume model covering the simulation of a single plume. Furthermore, information is needed whether the model is run online within the driving model or – driven offline. What is the time step of Cocip and the time step of the update of weather information (if Cocip is run offline). Does the model cover any interaction with the background fields?

Citations:

- In line 28 at least two of the citations have nothing to do with the size of the climate impact when conducting 'contrail avoidance' but talk about the dimensions of ISS. Please cite in a way that the citations stand next to the statements that they are supposed to support.
- In line 179, citation 33 – the longish ECMWF report does not talk about the accuracy of humidity forecasts. I could find only a mention of 'relative humidity' within the report that is mentioned twice in a table but does not appear to be discussed within the paper.
- Citation 22 appears to be a conference talk and not a publication.
- When citing webpages, it is important that at least the link to the page works.
- Several citations are missing the journal name or issue numbers.
- In general, DOIs appear not to work.

Data Availability: why are the model simulations not available

Reviewer #3

(Remarks to the Author)

As we mentioned in our initial review, we believe this research makes a significant contribution to the field and is, to our knowledge, one of the first studies to use sophisticated statistical techniques to address the inherent difficulties of assessing contrail avoidance success.

The main contribution of the paper, in our view, is the innovative experimental design (essentially a 'matching' method) introduced here to limit the impact of confounding factors (weather conditions) that reduce the power of null hypothesis testing in this context. This experimental design imposed significant constraints in terms of flight conditions and avoidance strategies (very close pairs of outbound and return flights), which clearly limited the scope of the study, but also provided clear statistical advantages. In our opinion, it will be quite difficult to achieve such control of confounding factors in more general flight/atm conditions in subsequent work. Given the small sample size (another limitation of the study), we also think that the use of non-parametric tests is perfectly sound from a statistical point of view.

Thanks to the reviewers' comments, the revised manuscript highlights more clearly the intrinsic limitations of the study, as the authors have removed some claims not necessarily related to the novel method developed here (significant uncertainties still surrounding the climate impact of contrails, intrinsic limitations of satellite observations...). The additional details on the use of cocip and the design of the ML algorithm are also clear improvements over the original manuscript.

The authors also replied to my question about the sample selection process (application of a subsequent CLZ prediction filter), which may raise questions about the prediction horizon and the avoidance strategy itself in future studies but does not change the outcome of the statistical analysis here.

In conclusion, the experimental method presented here is novel and addresses the structural problems inherent in attempting to statistically confirm the success of contrail avoidance strategies. The positive results certainly need to be confirmed by further research with significantly more flights and in more general configurations, both in terms of flight types, avoidance mechanisms and interaction with air traffic control. This seems to be the intention of the authors and is now clearly stated in the discussion part of the revised manuscript.

Reviewer #4

(Remarks to the Author)

Thank you for the revised manuscript.

This reviewer approves the publication of the revised manuscript pending one minor revision.

The use of the "internet" as a ground-to-air communication is not correct.

The definition of "internet" is as follows: "The internet is a global network of computers, servers, phones, and smart devices that are connected and can communicate with each other. It uses a packet routing network that follows the Internet Protocol (IP) and Transmission Control Protocol (TCP) to transmit data and media. The internet allows users to access data from other systems and interact with other users, and it provides services such as World Wide Web sites and data archives."

There is no ground-to-air communication that fits this definition.

Please update the manuscript to correctly communicate transmission channels. In the real world these channels have significant bandwidth limitations and have transmission costs.

Reviewers' comments:

Reviewer #1 (Remarks to the Author):

The paper deals with a very interesting topic which attracts high attention currently. The paper is written well and the overall aims of the work performed are reasonably described.

However, I recommend major revisions of the manuscript comprising in particular the conclusion section, because they are critical for the scientific content and technical soundness.

1. Is the paper technically sound?

Overall concept of alternative flight trajectory planning is a well-established method in the existing literature. Using satellite imagery for contrail detection is also well established. Similarly, combination of both for evaluating success of contrail avoidance strategies during real flights is also a well-established methodology and has been explored, implemented and demonstrated in other studies, e.g. Sausen et al., 2023 and Hofer et al., 2024. Overall, the paper is technically sound in describing its overall approach, however we recommend further clarity in terms of the statistics issue with such a small sample of flights – only 22 flights each group, in the light of the much to generalized conclusions drawn.

We thank the reviewer for the feedback, we acknowledge that we used a small number of flights (22 per group). We address this in the manuscript, highlighting the potential for increased statistical power and robustness with larger-scale trials. It is important to highlight however that with the right rigorous randomized controlled trial design to adjust for confounders - and the appropriate hypothesis testing for small sample sizes - we are able to show statistically significant results. Please see the response to question 4 for further detail on the statistical analysis.

2. Are the claims convincing? If not, what further evidence is needed?

The work presents a workflow how aircraft could avoid CLZ regions relying on information as forecasted by models. It is recommended that authors stick to this clear description, and hence

avoid to state that they are avoiding contrails (e.g. line 142), and avoid to describe displays while stating that no contrails were created (i.e. line 109), instead of indicating that they simply those regions where CLZ were forecasted were avoided. From the description I get the impression, that no validation or other confirmation of the real conditions, e.g. in terms of “skill of the forecast” or similar was performed. Hence, the authors of the study did not explore to what extent they were not able to observe an contrail, because in the real atmosphere no CLZ region existed. They count the non-existence as a success

We think that the reviewer may have misunderstood this element of our paper. For each flight (whether in the control or treatment group), we use satellite observations to assess whether a contrail was formed by that flight (see subsection "Satellite Image-Based Verification" in the "Experiment Design" section), and find that there are significantly more contrails detected in the control group (see subsection "Ascent/Descent Adjustments Lead to a Decrease in Detectable Contrail Formation" of the Results section).

3. Are the claims fully supported by the experimental data?

As result from my review, I come to the end, that the claims are not supported, critical points are not adequately assessed. By way of example, e.g. no methodology has been proposed in order to identify false forecast information, which could lead to wrong conclusions. To illustrate, the fact that no contrail is formed, could also be due to a wrong forecast of CLZ; and not as concluded by the authors, on a successful avoidance strategy. Additionally, the fail to mention that their estimates when quantifying climate effect of an avoided contrail relies only on parameterized numerical simulations.

Similar to the previous comment, we disagree with the referee’s statement that “no methodology has been proposed to identify false forecast information”. We used satellite observations to do this. For example, we found that in the control group, where all flights are predicted to make contrails, only 11 out of 22 flights did so. This is precisely an assessment of the accuracy of the forecast.

When the reviewer says “ the fact that no contrail is formed, could also be due to a wrong forecast of CLZ; and not as concluded by the authors, on a successful avoidance strategy”, this issue is exactly why we performed a randomized controlled trial. In the treatment group we observed 4/22 flights making contrails. If we had not used a control group, we would not know if this low fraction was due to forecast errors or because we avoided the CLZ. However, we are able to compare to a control group, which had similar conditions as the treatment group except

that the planes avoided CLZs. The fact that the control group created significantly more contrails indicates that contrail avoidance caused the difference.

Furthermore, I suggest major revision of the claims, more specifically:

Line 164f: The authors conclude that they have delivered a proof of concept for an avoidance of contrails and that they have shown that contrail avoidance is a viable strategy to combat climate change. To my understanding this is not supported.

We agree with the reviewer that this study has not demonstrated evidence of climate change impact, and we have removed the statement.

Line 168f: Further, they claim that this study is one of the first steps towards developing a comprehensive avoidance strategy. In the light of previous literature, this is seen to be kind of misleading. It is probably more adequate to consider this study as one further step towards developing a comprehensive avoidance strategy, more specifically they have completed an initial experiment on tactical contrail avoidance.

We have amended the text to say that this study is an important step towards developing comprehensive contrail avoidance.

4. Is the statistical analysis of the data sound?

Line 234ff: The authors are presenting results from quite a small sample size, i.e. 22 flights in each group. However, I have the impression that they fall short on proposing an adequate methodology, in order to proof the statistical significance of their results, which they have achieved with such a small sample.

Statistical significance is influenced by sample size, however it's also heavily dependent on the signal-to-noise ratio within the data [1]. Our crossover design trial mitigated noise by ensuring that the same aircraft, pilot, and approximately similar CLZ were present in both treatment and control groups, thereby controlling for potential weather-related confounders [2]. Given the small sample size, we avoided asymptotic assumptions about normality and adopted a non-parametric approach, enhancing the statistical power of our hypothesis test in this context [3, 4].

[1] Sullivan GM, Feinn R. Using Effect Size-or Why the P Value Is Not Enough. J Grad Med Educ. 2012 Sep;4(3):279-82. doi: 10.4300/JGME-D-12-00156.1. PMID: 23997866; PMCID: PMC3444174.

[2] Senn, S. (2013). Cross-over trials in clinical research. John Wiley & Sons.

[3] Wilcoxon, F. Individual comparisons by ranking methods. Biom. Bull.1, 80–83 (1945)

[4] Pratt, J. W. Remarks on zeros and ties in the Wilcoxon signed rank procedures. J. Am. Stat. Assoc.54, 655–667 (1959).

5. Does the availability of data adhere to the expected standards of your research community?

Access to data

Line 321 In the manuscript the explanation “The datasets used and/or analyzed during the current study are available from the corresponding offer on reasonable request.”, to my understanding does not make sense. It should be probably noted author here, instead of offer.

This was indeed a typo, thank you for noticing. We have fixed it to read ‘author’.

However, it is strongly suggested to upload the data to a repository, e.g. zenodo, or similar, assuring reproducibility of this study.

Please see the supplemental material repository (link) which has a table with flight-level information, links to all flightpath visualizations used by evaluators, as well as the colab used for the statistical analysis. All have been made publicly available.

6. Are the claims appropriately discussed in the context of the previous literature?

Line 42: Early literature is not taken into account, e.g. Hermann Mannstein, Peter Spichtinger, Klaus Gierens, A note on how to avoid contrail cirrus, Transportation Research Part D: Transport and Environment, Volume 10, Issue 5, 2005, Pages 421-426, ISSN 1361-9209, <https://doi.org/10.1016/j.trd.2005.04.012>. (<https://www.sciencedirect.com/science/article/pii/S136192090500026X>), but also Green et al. 2003 is not mentioned.

We thank the reviewer for bringing these early references to our attention, we have added them.

Line 28: Using the Lee et al. 2021 is not adequate for providing reference on an “extremely cost-effective way to reduce anthropogenic climate forcing”, hence this reference needs to be deleted here.

We have deleted this reference.

7. If the manuscript is unacceptable in its present form, does the study seem sufficiently promising that the authors should be encouraged to consider a resubmission in the future?

Major revisions are suggested.

8. Is the manuscript clearly written? If not, how could it be made more accessible?

Yes, the manuscript is clearly written.

9. Are there any special ethical concerns arising from the use of animals or human subjects?

Not applicable.

Reviewer #2 (Remarks to the Author):

Comment 1:

The authors report on a study that tests the feasibility of contrail avoidance by flight rerouting, a topic that has gained much attention lately. The question being tested is if ice supersaturated regions can be forecast with enough precision to be able to reroute flights in order to avoid contrail formation. The authors study 11 ice supersaturated areas, for which the forecast of a ML algorithm and a physically based model agree, reroute flights crossing those ice supersaturated areas and use satellite observations to study whether the flight rerouting was successful. I appreciate the effort that went into this work but the study makes a number of assumptions that limit the significance of their work.

We thank the reviewer for the feedback, and would like to clarify that we used 22 distinct CLZs that were encountered by the 44 flights in our study.

Comment 2:

The authors are careful to write in the main text that they check for 'detectable' contrails, in large parts of the paper the word 'detectable' is missing and instead the authors write about 'contrail avoidance' instead of 'avoidance of detectable contrails'. This is not only a wording but a science issue. If the authors write that they avoid contrails then they suggest that the whole climate impact connected with the contrail has been avoided. If, on the other hand, it is simply the contrail's optical depth that has been reduced to below the detection threshold then only a part of the contrail climate impact has been avoided. Since contrails are typically optically very thin clouds, the difference in terms of optical depth may not even be large. Furthermore, it may be that the contrail's ice water content and with it the optical depth increases more slowly after rerouting so that they become visible after the maximum 40 min., the maximum time delay allowed in this study. Differences in contrail optical depth and its development may be connected with differences in the atmospheric conditions at the lower flight level relative to the higher flight levels that control contrail ice nucleation and water vapor deposition. Therefore, by flying lower it may either be that contrails do not form or that contrails form but with changed optical depth which may make them not detectable or detectable only later in their life cycle. In any case, 'contrail avoidance' cannot be shown, at least not with the methods used here.

Response:

We have revised the wording to be more precise by using “detectable contrail avoidance” among other similar phrases throughout the paper. We have also added paragraphs to the introduction and discussion section highlighting the difference between the formation of contrails and the formation of detectable contrails. Finally we note that while the satellite image evaluators used a guideline that first detectable onset of a contrail is typically before 40 minutes, the evaluators were able to take into account 2 hours of satellite imagery.

Comment 3:

The motivation of this work is the assumption that contrail avoidance is a promising climate change mitigation strategy. This has never been shown conclusively. It is well known that the simulation of clouds is the largest obstacle in climate modelling and weather forecasting. The uncertainty connected with cloud properties and relative humidity is large and translates into a large uncertainty in contrail properties and their climate impact. The present paper shows that the avoidance of detectable contrails can be achieved by a ‘mere’ 2% increase in fuel per adjusted flight. This additionally emitted CO₂ will have an impact on radiation for centuries and the gain will be the avoidance of a short-lived climate impact of uncertain size that is connected with the avoidance of detectable contrails. But of course, in some (or many?) cases, rerouting does not have the desired outcome and the 2% increase in CO₂ emissions may be in vain. Increasing the climate impact of long-lived climate forcers in order to reduce the impact of short-lived climate forcers of uncertain impact is potentially very dangerous. The fact that not all climate change components connected with aviation, that may also change due to rerouting, have been quantified complicates the problem further. This means that the present paper studies the feasibility of a mitigation option for which the resulting change in the climate impact of aviation is currently impossible to robustly quantify.

Response:

We thank the reviewer for bringing to our attention that the modeling complexities to measure the impact of contrail avoidance as a climate change mitigation bring a lot of uncertainty into the conclusions. We agree with the reviewer that there are many uncertainties around contrail avoidance as a climate change mitigation strategy. As the reviewer points out, one of these uncertainties is whether, and how often, contrails can be successfully avoided. We think our work, which attempts to avoid satellite-detectable contrails and measure whether that avoidance

was successful, is an important contribution to reducing this uncertainty. We agree that uncertainties in cloud modeling and contrail properties makes estimating the climate impact of contrails challenging especially when considering the additional CO₂ emissions from rerouting. However these questions are outside the scope of this paper. In the abstract we have replaced “Contrail avoidance is a promising climate change mitigation strategy” with “Contrail avoidance is a potential option to mitigate this warming effect”.

Comment 4:

Figure 2 shows that from a large number of flights only very few form detectable contrails even though the length of the detectable contrails formed would indicate a larger ISSR. Even in the vicinity of those detected contrails other flights appear not to form a detectable contrail. Detectability of contrails depends on many variables. A larger fuel use leads to an increased contrail optical depth which makes it easier to detect the contrail. A higher altitude leads to a larger temperature difference between surface and contrail which impacts the signal in OLR. Many of the detected contrails end in the area with natural cloudiness over an island. Lower cloudiness and changing surface conditions are known to make contrail detection more difficult. It is very likely that many more contrails formed but were or could not be detected. The failure to detect a contrail does not mean that the contrail is not there or has no climate impact. This supports the earlier argument that the mere fact that a contrail cannot be detected does not mean that it has been avoided.

Response:

We appreciate the reviewer's observations regarding Figure 2 and the challenges of contrail detectability. We agree that various factors, including optical depth, altitude, cloudiness, and surface conditions, can influence contrail visibility in satellite imagery.

However, it's important to note that the flight paths shown in Figure 2 are at substantially different altitudes. While multiple flight paths might appear in a single scene, this altitude variability significantly reduces the likelihood of many of them intersecting an ISSR. Our CLZ predictions, as detailed in the "Flight Sample Inclusion & Exclusion Criteria" subsection of the "Experiment Design" section, are inherently altitude-specific, ensuring we target flights with a

higher probability of contrail formation. Consequently, because many flight paths in Figure 2 were unlikely to have encountered the specific altitude range of the predicted CLZ(s), we do not necessarily expect detectable contrail formation rates to be high just because they appear close in the 2D projection.

Comment 5 & 6:

There is next to no model description for CoCip other than it is a 'physics based' model. Please expand.

How are CLZs calculated within CoCip?

Response:

Thank you for pointing this out. We have extended our description of CoCiP in the Methods section of the revised manuscript.

Comment 7:

Your neural network approach uses many input variables. Which variables are most important for the successful prediction of contrail formation? Which variables are not important? Does your analysis agree with earlier studies?

Response:

We have done an ablation study and have added the details in response to this question in the manuscript's Methods section.

Comment 8:

By choosing ice supersaturated areas for which two very different forecasts agree, the authors are picking situations for which the predictability is high and the ice supersaturated areas are likely very large. This means that the authors are not randomly sampling but choosing systems for which their approach is most likely to work. Therefore, the results of this study should not be interpreted as proof of concept after which only upscaling is required.

Response:

We acknowledge that requiring agreement between the ML and CoCiP models decreased the likelihood of including false positive CLZs in the study. In the discussion we go over several factors that need to be explored in future trials which the limited generalizability of our study did not cover, such as the limited sample size, the focus on specific avoidance adjustments (ascent/descent) and geographic regions, among others. We further expanded this discussion in the revised manuscript to mention the need to relax the requirement for agreement between ML and CoCiP models in CLZ identification.

Comment 8b:

You write that 'ISSRs are rare and cover a low portion of the upper troposphere. Therefore, only a small percentage of flights would need to make small adjustments to avoid the majority of contrail warming': please be more precise and give numbers and discuss how certain those statements are.

For the ISSR fraction, we've added an additional reference (Gierens 1999) where the ISSR fraction is computed from in-situ data, and we've included the fractions they found in the text. For the percentage of flights needed to make small adjustments, we have included the range of numbers found in the cited references.

Comment 9:

What are the predicted spatial and temporal scales of the ISSRs you are trying to avoid?

Contrails absorb and reemit outgoing longwave radiation.

Response:

We have added the spatial scales, with appropriate references, to the text. We are not aware of any reference that quantifies the temporal scales of the ISSRs, this is a difficult question to answer since the regions move and change shape over time. Our own observations of the weather data indicate that the ISSR regions seem to exist for several hours.

Comment 10:

Line 91: please give pressure and temperature at flight level and not just FL320.

Response:

We have included this information in the revised manuscript.

Comment 11:

Line 143-144: the true positive could also be due to attributing the contrail to a wrong flight.

Response:

We agree with the reviewer that the attribution of contrails to the wrong flight is a potential source of true positives. However, the structure of the crossover trial is designed so that the probability of wrong attribution is balanced between the two groups. Therefore, while this might introduce some noise into the data, it should not systematically bias the overall conclusions of the trial.

Comment 12:

Lines 164-168 are simply claims not supported by any evidence within this paper. If the authors want to claim that the contrail warming is larger than the warming due to the additionally emitted CO₂ then they should include an estimate of the contrail warming and its uncertainty and estimate the overall climate impact of the flight with/without rerouting. Furthermore, they should include a description of the method how they are calculating all climate impact components associated with the flight and discuss the uncertainties.

Response:

We agree with the reviewer's feedback regarding the claims made in lines 164-168. We acknowledge that the current study does not provide direct evidence to support the comparison between the warming impact of additional fuel burn and the avoided contrail warming. We have removed this statement from the manuscript.

Reviewer #3 (Remarks to the Author):

Comment:

The paper examines the key challenge of the feasibility of contrail avoidance in commercial aviation. The published results are an important contribution to the topic and show that efforts in this research area should be continued to confirm the positive outcome on a larger scale and to include more elements of the air traffic control network.

The main claim of the paper is that it is operationally feasible for commercial flights to avoid contrail formation on a per-flight basis by making altitude adjustments based on predicted contrail-likely zones (CLZs). This claim is based on the use of prediction tools developed by the authors in previous research, namely a combination of a physics-based simulation model and a machine learning model to perform the prediction task. These predictions guide pilots and air traffic control in making altitude adjustments to avoid these zones in both the strategic and tactical phases. To focus on the feasibility of the avoidance strategies on a flight-by-flight basis, the authors chose to focus on conditions where full air traffic control was not required, thus simplifying both the operational method and the statistical analysis.

The test results were performed by a rigorous randomised controlled trial (RCT) comparing the frequency of occurrence of contrails for a set of flights diverted to avoid CLZs (the treatment group) with the frequency of a control group. The main difficulty with the statistical analysis comes from the presence of confounding factors (weather conditions), the small sample size (22 flight pairs) and the fundamentally counterfactual nature of the null hypothesis being tested. To mitigate this difficulty, the authors decided, in a very pragmatic way, to focus on pairs of very close outbound and return flights, to assign the outbound flight to a group and the return flight to the opposite group, thus trying to reduce the role of confounding factors.

From an experimental design point of view, we believe that the 'matching' method used here is particularly clever and effective in reducing the role of confounding factors, especially given the small sample size (22 pairs of control/treatment flights). The risk obviously lies in the change in weather conditions during the 2 hours between the outward and return flights. In future work

with significantly more flights, it might be interesting to compare other approaches to mitigate these effects (stratification, regression-based statistical adjustments...).

From a statistical point of view, the use of non-parametric tests is fully justified here given the small sample size (Bayesian approaches were also possible), increasing the test statistical power and confidence in the final p-value. However, again, a drastic increase in sample size in future work would be the only way to really confirm the results.

Response:

We thank the reviewer for the thoughtful feedback. We agree that the potential for changing weather conditions between outbound and return flights is a key consideration in this trial design. We are encouraged that, despite this limitation, the crossover design chosen to control for confounders together with the non-parametric hypothesis test was sufficient to detect a significant treatment effect. However, we agree that future trials with a considerably larger number of flights are necessary to solidify these results.

Comment:

Although the experimental design is perfectly described in the paper, we would like to clarify our understanding of the sample construction. We understand that the selected flights (whose trajectory crossed a CLZ) were selected based on a prediction two days before the flight, but that the avoidance strategy was adjusted on the day of the flight. Does this mean that the 22 CLZs predicted two days before were confirmed by a new prediction two days later (which seems to contradict slightly the 50% precision in the control group), or are the 22 flights the result of a subsequent filter? However, this question is not crucial as the article does not formally address the issue of the prediction horizon.

Response:

We thank the reviewer for raising this important question regarding the sample construction and CLZ prediction process. The 22 treatment and control pairs of flights are the result of a subsequent day-of CLZ forecast filter being applied.

Comment:

In conclusion, this paper is an important contribution to the field, with plausible and positive results that need to be confirmed in subsequent research with more flights and a more complete air traffic control network. The statistical methodology and experimental design are particularly well suited to the difficult task at hand (statistical hypothesis testing with confounding factors and a small sample size).

Response:

We thank the reviewer for their thorough and positive assessment of our work and its contribution. We are encouraged that they recognized the appropriateness of our statistical methods and experimental design. We absolutely agree that future research with larger sample sizes is essential to confirm and build upon these findings.

Reviewer #4 (Remarks to the Author):

Thank you for the opportunity to review this excellent paper. The design of experiment, the methods/tools used, and the results are all contributions to the literature and advancing this important work.

Please accept the following minor comments;

(1) Abstract/Intro; refers to "major" "significant" and "substantial" contribution of contrails to total anthropogenic impact. This seems to be overstating the case given that estimate of contrails contribution is 2% (Lee et.al. 2021). Why not just say 2%

The reason not to use the 2% number is that it is a comparison of radiative forcings, and so doesn't take into account the difference between CO₂ (which lasts a long time) and contrails which are temporary. This is not the subject of this experiment and paper so we don't want to highlight it in the first sentence of our abstract. We have changed the first sentence to read that contrails are a major component of aviation's contribution to climate change.

(2) Intro, para 3 "often struggle" Is this a scientific term? Perhaps "subject to inaccuracies" and reference Gierens, Geraedts. handen, Reutter,...

Response;

We thank the reviewer for bringing this to our attention, we have changed the wording as suggested and have added the relevant references.

(3) Intro, para 3 "navigational contrail avoidance" term is used here. This is a key concept. What is it?

We have defined this term in the text.

(4) Intro, Para 3. This para discusses the challenges of implementing Contrail Avoidance (i.e. predicting ISSR and airline systems). Between these two is mentioned of the difficulty designing an experiment (line 33, 34). This is confusing.

We have moved the sentence about the experiment design to the end of the paragraph.

(5) Intro, Para 4. Discusses Saussen "Complete air traffic control" What do the authors mean by this. As a pilot in controlled airspace, the flight is always under complete air traffic control.

Perhaps what is being referred to is that ATC determined the CLZ and directed the flights away from them. i.e. this was all done by ATC (not the airline).

Changed to “control of all flights in a region”. The reviewer is correct that the Sausen trial requires the ability (which ATC has) to control all flights in a region, whereas the current work does not.

(6) Intro Section. There are two paragraphs buried in the Discussion and Conclusion section paras 4 & 5, that do nice job in explaining the motivation for this study. It extends/complements the Sausen et al study by using only airline ops independent of ATC. These 2 paras are well written.

We have taken the reviewer’s suggestion and moved these paragraphs into the introduction.

(7) Intro identifies a "physics-based model" Why not identify this as CoCIP and credit Schumann. Also explain that the (atmospheric) physics-based model uses atmospheric forecast data on the flight-path to predict whether Contrail will occur.

We have identified the model as CoCip and referenced Schuman in the introduction, we have also expanded on what CoCip is and go into detail on how we use it in the first paragraph of the Methods Section.

(8) Intro identifies a "machine learning" model. This is very vague. Perhaps explain the ML model supplements the CoCip atmospheric physics-based model. See description in Methods section para 2 and 3.

We would prefer not to move the whole description from the Methods section to the introduction, but we have changed the introduction text to state that the ML model is designed to correct the weather forecast data. We have also added a reference to the methods section.

(9) Intro, last para, 2nd sentence. Confound the purpose of the study (i.e. per flight basis) with the detail of the design of experiment (i.e. blind human raters).

Changed to: “By using (treatment- and control-group) blinded human evaluators and satellite imagery, we assessed contrail formation on a per-flight basis.”

(10) Results, para 1. "CLZ predictions over the internet" ??? Somewhere else in the paper is mentions ACARS and SATCOM. This is important, cause ACARS and SATCOM have limited bandwidth and are expensive.

We do not believe we have made any reference to ACARS/SATCOM in the manuscript. Using the PACE software, any internet connection (including a connection not using ACARS/SATCOM), is capable of transmitting CLZ information.

(11) Results, para 2. The discussion on "tactical near airport contrail avoidance" is VERY important. This is opposed to Cruise tactical flight level changes. Why was this approach used. This reviewer assumes that it was applied since it avoided amending the Cruise Flight Level that would require interaction with dispatch and air traffic control. Is this hypothesis correct.

This approach requires flying at lower than planned altitudes, resulting in additional fuel burn. A cruise flight level change may reduce the additional fuel burn. This should be explained.

We opted for a tactical near-airport contrail avoidance intervention for several key reasons: 1) This approach allowed for an effective crossover trial design, as we had flexibility in selecting destination airports near CLZs. This minimized the time difference between outbound and inbound flights encountering the same CLZ, thus controlling for potential confounders like aircraft type and atmospheric conditions. 2) Near-airport maneuvers eliminated the need for additional ascents or descents during cruise, simplifying operations and ensuring higher pilot compliance. 3) Compared to modifying cruise levels, which could require prolonged low-altitude flight or multiple altitude changes, delayed ascent/early descent minimized additional fuel usage by avoiding extra climbing. Additionally, lower altitudes are associated with a reduced probability of contrail formation even in the presence of some ice supersaturation. Finally, it is important to note that all flight path changes, including near airport avoidance, require communication with ATC and dispatch.

(12) Results, para 4, Add nautical miles 8km (4.3 nm) and 13 km (7 nm). These are very short contrails? No?

This sentence was unclear; we are stating the altitude range for which we produced contrail forecasts. The text has been updated to indicate this.

(13) Results, para 4, Is there any data that can be reported on their duration (e.g. 2 hours).

We assume the reviewer is referring to the duration of the CLZ predictions, and we have amended the text to indicate that the CLZs were updated hourly.

(14) Results, para 4. How long do ISSR last? Is two hours short enough that the ISSR will still be there?

It is likely that some ISSRs would not remain over the turnaround airport for long enough to have both legs passing through ISSRs. We have modified the text to explain that we only considered cases where both legs of the flight pair were forecast to pass through a CLZ.

It is also worth noting, that the probability of the ISSR not being present on the turnaround leg is balanced between the treatment and control groups due to the randomization of the crossover trial design. Therefore, while this might introduce some noise into the data, it should not systematically bias the overall conclusions of the trial.

(15) Results, CLZ Avoidance Planning and Execution, para 2 is this "clear air turbulence" or just turbulence

Clear air turbulence; we have updated the text accordingly.

(16) Results, Sat-Image Verification, para 2. How does the satellite image show the "exhaust plume" What is the satellite image actually showing.

Apologies for the use of the term "exhaust plume," we have now standardized the manuscript to use the phrase "(advected) flight trajectory" to be more clear.

The satellite images are false-color infrared images designed to highlight the existence of clouds. In particular thin cirrus clouds should show as dark features (we have amended the text to explain this). In such an image contrail cirrus are dark lines, and we are assessing whether these dark lines are lining up with the expected position of the advected flight trajectory.

(17) Results, Sat-Image Verification, para 2. Last few sentences are confusing. What are authors trying to say.

We have amended the text to clarify that we are describing the labeller's use of the automated contrail detections. At the time of the study, the automated system was not as good as an expert human at finding contrails in the false color images, so the determination of whether a contrail existed was ultimately made by the expert human evaluators. Nevertheless the automated detections were a valuable tool to help the labelers.

(18) Results, Ascent/Descent Adjustments, para 2, 1st sentence. Confusing. Why is this metric useful. Contrails contribute to <2% of the stage-length. The treatment was approx 1%. So the treatment reduce the contrail length by 50%. Is this correct?

Correct, this 54.4% is the result of dividing the contrail length percentages in the treatment and control groups.

(19) Results, Ascent/Descent Adjustments, para 2, last sentence. How was this estimate derived. Please show equations/calculations. Also 10.6 kg?

This number in Kg of CO₂e, it is the mean contrail warming estimate of 57.4 mW/m² from Lee et al 2021, divided by the total number of flight km from 2018 in that same work, and then converted to a ton of CO₂e using GWP100. These data comes from the spreadsheet of the supplementary material in the cited work. We added our calculations in the supplementary material.

(20) This reviewer does not understand why the Methods section appears after the Discussion and Conclusions section. This information would have been useful in the Results Section.

Our understanding is that this is the format preferred by the journal.

(21) Also, the Results section is really a description on of the Methodology (i.e. Design of Experiment). The design of Experiment is unique contribution to the literature and should be described clearly, such that future papers can refer to this.

We thank the reviewer for the suggestion and have expanded and moved the “Experiment Design Section” after the Introduction.

(22) What is the Conclusions of the study? What are the limitations of this study? What is future work? State these clearly independent of the Discussion.

Thank you for bringing to our attention that this is not clear in the manuscript, we have split the Discussion from the Conclusions section in order to clarify this.

Reviewers' comments:

Reviewer #2 (Remarks to the Author):

The paper studies the feasibility of contrail avoidance. It concentrates on operational issues avoiding CLZs (contrail likely zones) and on the associated changes in satellite-based contrail detection. I do not have a problem with the main work and agree with the authors that the size of the data set is rather limited. Additionally, the paper makes some unsupported claims about the climate impact of this mitigation option that need to be removed.

The authors say in line 43-45 'We focused exclusively on demonstrating the feasibility of contrail avoidance on a per-flight basis and did not consider the radiative forcing of avoided contrails'. This appears to be a fair description of their work and, therefore, any claims that 'the work suggests that contrail avoidance could lead to a significant reduction in the climate impact of aviation' (last sentence of the abstract) are unsupported and should be cut.

We have removed the part of the sentence the reviewer requested.

The authors frame their work by mentioning only 'uncertainty surrounding operational constraints and accurate formation predictions' (2nd sentence abstract) as problems connected with proposing contrail avoidance as a mitigation option. This subject is expanded on in the paragraph starting in line 33. In line 174-175 it is simply stated that 'A larger trial with more statistical power could also quantify the amount of warming prevented through contrail avoidance, allowing estimates of the cost-effectiveness of contrail avoidance ...'. This seems to imply that it is straightforward to estimate the climate impact of contrail cirrus and that estimates do not suffer from uncertainties. The conclusions finish by saying that the authors 'hope that their study motivates airlines and policy makers to avoid contrails'. Even though I understand that the climate impact of 'contrail avoidance' is not the subject of this paper, it is necessary to relate the topic of this paper to the bigger picture. I have seen no discussion about the problems and uncertainty when estimating the climate impact of contrails and contrail avoidance, when comparing long-lived and short-lived forcers..... In the contrary, the authors appear to suggest that the 'quantification' of the climate impact is straightforward. This is despite the fact that some relevant papers are cited. The two Lee et al papers describe a large number of uncertainties and can lead the authors to papers describing uncertainties in more detail. Finally, the authors failed to acknowledge the uncertainty coming from the fact that contrails may not have been avoided but that their optical depth may be decreased so that they are not detectable any longer while still having a climate impact (see my last review).

In the previous submission, we updated the manuscript to clarify in every instance that this study is discussing a decrease in the frequency of detectable contrails. The climate impact of detectable vs undetectable contrails is outside the scope of this work, so to address the reviewer's concern, in this revision we have reworded the last sentence of the conclusion, and no longer mention policymakers or airlines. We also removed the

following statements from the manuscript (formerly in line 174 of the first resubmitted version):

A larger trial with more statistical power could also quantify the amount of warming prevented through contrail avoidance, allowing estimates of the cost-effectiveness of contrail avoidance which are essential for comparing broad-scale contrail avoidance to other sustainability initiatives.”

Overall, the paper strengthens a growing body of literature of the feasibility of contrail avoidance. I don't see the paper as an important step forwards since the main road block is how to quantify the climate impact of the mitigation option.

Additional major comments:

Line 27-28: From the fact that ISS can be found in 13.5% of cases in the upper troposphere and 2% in the stratosphere you cannot conclude that between 2% and 13,5% of flights form a contrail. After all, during a flight, contrails will form only in a small fraction of the flight distance but the number of affected flights can be large. Furthermore, all flights that are in the stratosphere need to have passed through the troposphere so that a 2% probability of contrail formation is significantly too low. Additionally, a newer paper from Petzold et al 2020 looks at in-situ IAGOS data and finds that over all of the analyzed areas (Europe, North Atlantic and eastern North America) 'air masses close to the tropopause level are nearly saturated with respect to ice and contain a significant fraction of ISSRs with a distinct seasonal cycle of minimum values in summer (30 % over the ocean, 20 %–25 % over land) and maximum values in late winter (35 %–40 % over both land and ocean).' The authors need to update ISS frequencies and need to acknowledge that more flights need to be diverted in order to reduce the contrail climate impact.

Apologies for our imprecise language mentioning only ISSR in this sentence; we have corrected it to refer more generally to the atmospheric conditions along flight paths that are estimated to cause the majority of contrail warming, supported by the quantifications found in Refs 7-11. Unfortunately the reviewer's recommended reference (Petzold et al 2020) only analyzes prevalence of ISSR (and not Schmidt-Appleman criteria or estimated radiative forcing of contrails), so it does not directly apply here.

Line 53: The following statement is wrong 'Simulations of navigational contrail avoidance find that avoiding ISSR regions prevent contrails from forming'. It is well known that ISS is not needed for contrail formation – it is only needed for contrail persistence. Furthermore, knowledge of contrail formation conditions is not a result of simulations conducted in connection with contrail avoidance. This is based on our understanding of cloud physics and supported by observations. There are a number of papers by Schumann et al. on this topic, e.g. the Schumann 2005 which you cite already.

Line 143: You say that in around 1-2% of total flight km a contrail was formed. This value is surprisingly low even if a large part of the flight km were flown within the stratosphere - which

appears not to be the case in your figure 1. Since temperatures in most of the northern hemispheric main air traffic areas are such that contrail formation conditions are met when air is ice supersaturated, this means that the formation probability of detectable and non-detectable contrails is much larger than what you estimate. Apparently, you are only observing the contrails with the largest optical depth and missing the bulk of contrails which appear to remain non-detectable (see Kärcher et al. 2009 'Factors controlling contrail cirrus optical depth'). It is important to acknowledge the existence of a large number of non-detectable contrails as they explain part of the climate impact of contrail cirrus.

We agree with the reviewer, in particular we should have said 'prevent persistent contrails' rather than 'prevent contrails.' We have reworded the sentence to reflect this.

In the previous revision of the manuscript submitted in August (in response to the reviewer's point about detectable contrails) we added two additional paragraphs to the manuscript to emphasize that our work's scope is limited to satellite-detectable contrails: an Introduction and a Discussion paragraph starting in line 52 and 191 respectively. In that revision, we also made it clear in every instance that this study is discussing a decrease in the frequency of detectable contrails. The climate impact of detectable vs undetectable contrails is outside the scope of this work.

Line 148-149: In the supplementary material you give details about your calculation of the contrail cirrus equivalent CO₂ emissions. You divide contrail cirrus ERF by the AGWP of CO₂. This calculation does not give the equivalent CO₂ emissions connected with contrail cirrus. You should have divided the AGWP of contrail cirrus with that one of CO₂ in order to get equivalent CO₂ emissions. Please note that in several places in the supplementary material units are incorrect or missing.

We removed the last sentence of the Results section (sentence that started in line 152 of the version submitted in August) which provided the CO₂ equivalence estimate as this is out of the scope of our study.

Line 154-155 and 161-162: by targeting the same CLZ with outbound and inbound flights you may not eliminate the problems connected with changing atmospheric conditions. Moisture fields are very small scale and can significantly vary on small spatial and short temporal scales.

By targeting the same CLZ we reduced the change in atmospheric conditions. We did not mean to give the impression that we have eliminated such differences, and we have modified both sentences referenced by the reviewer to make this clearer to future readers.

The model description does not yet cover all the necessary information, e.g. it is important to say that Cocip is a plume model covering the simulation of a single plume. Furthermore, information is needed whether the model is run online within the driving model or – driven

offline. What is the time step of Cocip and the time step of the update of weather information (if Cocip is run offline). Does the model cover any interaction with the background fields?

We have added a paragraph to the Methods Section of the manuscript (line 208) providing the particular parameter details used to configure the model. We refer the reviewer and readers to our references for CoCiP's background field interaction details.

Citations:

- In line 28 at least two of the citations have nothing to do with the size of the climate impact when conducting 'contrail avoidance' but talk about the dimensions of ISS. Please cite in a way that the citations stand next to the statements that they are supposed to support.

- In line 179, citation 33 – the longish ECMWF report does not talk about the accuracy of humidity forecasts. I could find only a mention of 'relative humidity' within the report that is mentioned twice in a table but does not appear to be discussed within the paper.

We have removed this citation

- Citation 22 appears to be a conference talk and not a publication.

Citation 22 (20 in the resubmitted version of the paper) refers to a preprint available online: *Forecasting contrail climate forcing for flight planning and air traffic management applications: The CocipGrid model in pycontrails 0.51.0*. We have updated the reference.

- When citing webpages, it is important that at least the link to the page works.
- Several citations are missing the journal name or issue numbers.
- In general, DOIs appear not to work.

We have revised the links, DOIs and references in the manuscript. At the time of writing, we have confirmed that all links are functional. Please note that we cannot guarantee the future functionality of these links.

Data Availability: why are the model simulations not available

CoCiP simulations can be reproduced using the pycontrails API at api.contrails.org. The data availability statement has been updated accordingly.

Reviewer #3 (Remarks to the Author):

As we mentioned in our initial review, we believe this research makes a significant contribution to the field and is, to our knowledge, one of the first studies to use sophisticated statistical techniques to address the inherent difficulties of assessing contrail avoidance success.

The main contribution of the paper, in our view, is the innovative experimental design (essentially a 'matching' method) introduced here to limit the impact of confounding factors (weather conditions) that reduce the power of null hypothesis testing in this context. This experimental design imposed significant constraints in terms of flight conditions and avoidance strategies (very close pairs of outbound and return flights), which clearly limited the scope of the study, but also provided clear statistical advantages. In our opinion, it will be quite difficult to achieve such control of confounding factors in more general flight/atm conditions in subsequent work. Given the small sample size (another limitation of the study), we also think that the use of non-parametric tests is perfectly sound from a statistical point of view.

Thanks to the reviewers' comments, the revised manuscript highlights more clearly the intrinsic limitations of the study, as the authors have removed some claims not necessarily related to the novel method developed here (significant uncertainties still surrounding the climate impact of contrails, intrinsic limitations of satellite observations...). The additional details on the use of cocip and the design of the ML algorithm are also clear improvements over the original manuscript.

The authors also replied to my question about the sample selection process (application of a subsequent CLZ prediction filter), which may raise questions about the prediction horizon and the avoidance strategy itself in future studies but does not change the outcome of the statistical analysis here.

In conclusion, the experimental method presented here is novel and addresses the structural problems inherent in attempting to statistically confirm the success of contrail avoidance strategies. The positive results certainly need to be confirmed by further research with significantly more flights and in more general configurations, both in terms of flight types, avoidance mechanisms and interaction with air traffic control. This seems to be the intention of the authors and is now clearly stated in the discussion part of the revised manuscript.

Reviewer #4 (Remarks to the Author):

Thank you for the revised manuscript.

This reviewer approves the publication of the revised manuscript pending one minor revision.

The use of the "internet" as a ground-to-air communication is not correct.

The definition of "internet" is as follows: "The internet is a global network of computers, servers, phones, and smart devices that are connected and can communicate with each other. It uses a packet routing network that follows the Internet Protocol (IP) and Transmission Control Protocol (TCP) to transmit data and media. The internet allows users to access data from other systems and interact with other users, and it provides services such as World Wide Web sites and data archives."

There is no ground-to-air communication that fits this definition.

Please update the manuscript to correctly communicate transmission channels. In the real world these channels have significant bandwidth limitations and have transmission costs.

We thank the reviewer for pointing to this confusion. Many aircraft have on-board internet connections provided by vendor WiFi networks such as Viasat, Panasonic, and Go-go, which can work with a combination of ground-to-air and satellite systems. It is not uncommon for passengers to be able to access the internet during a flight. The PACE software uses the same connection, so we required the planes to have cockpit wifi to receive CLZ prediction updates. Note that the PACE software is running on a tablet, it is not directly integrated with the aircraft's systems. We have provided a summary of this in line 72 of the revised manuscript.